# A Systematic Review of Atypical Endometriosis-Associated Biomarkers

**DOI:** 10.3390/ijms23084425

**Published:** 2022-04-17

**Authors:** Ludovica Bartiromo, Matteo Schimberni, Roberta Villanacci, Giorgia Mangili, Stefano Ferrari, Jessica Ottolina, Noemi Salmeri, Carolina Dolci, Iacopo Tandoi, Massimo Candiani

**Affiliations:** Gynecology/Obstetrics Unit, IRCCS San Raffaele Scientific Institute, 20132 Milan, Italy; schimberni.matteo@hsr.it (M.S.); villanacci.roberta@hsr.it (R.V.); mangili.giorgia@hsr.it (G.M.); ferrari.stefano@hsr.it (S.F.); ottolina.jessica@hsr.it (J.O.); salmeri.noemi@hsr.it (N.S.); dolci.carolina@hsr.it (C.D.); tandoi.iacopo@hsr.it (I.T.); candiani.massimo@hsr.it (M.C.)

**Keywords:** endometriosis, atypical, atypical endometriosis, marker, biomarker, atypia

## Abstract

Ovarian endometriosis may increase the risk of malignancy. Several studies have suggested atypical endometriosis as the direct precursor of endometriosis-associated ovarian cancer. We performed an advanced, systematic search of the online medical databases PubMed and Medline. The search revealed *n* = 40 studies eligible for inclusion in this systematic review. Of these, *n* = 39 were finally included. The results from included studies are characterized by high heterogeneity, but some consistency has been found for altered expression in phosphoinositide 3-kinase (PI3K)/AKT/mTOR pathway, ARID1a, estrogen and progesterone receptors, transcriptional, nuclear, and growth factors in atypical endometriosis. Although many targets have been proposed as biomarkers for the presence of atypical endometriosis, none of them has such strong evidence to justify their systematic use in clinical practice, and they all need expensive molecular analyses. Further well-designed studies are needed to validate the evidence on available biomarkers and to investigate novel serum markers for atypical endometriosis.

## 1. Introduction

Endometriosis is a chronic, estrogen-dependent, progressive disease affecting approximately 10% of women of reproductive age [1]. Initially considered a benign disease, endometriosis, and particularly ovarian endometriosis (OMA), may increase the risk of developing malignancy. An association between endometriosis and ovarian cancer was initially proposed in 1925 by Sampson, describing endometrial carcinoma of the ovary arising in endometrial tissue [2]. Then, the transition from endometriosis to ovarian cancer was confirmed in 1953, when Scott wrote about malignant changes in endometriosis and pointed out that benign endometriosis might be observed in proximity to endometriosis-associated ovarian cancer (EAOC) [3].

It has been estimated that 0.5–1% of endometriosis cases are complicated by neoplasia, with a lifetime risk of about 1.9%, but it is relatively increased compared to the general population, having a lifetime risk of approximately 1.4% [4]. In a pooled meta-analysis of 13 case-control studies, the frequency of self-reported endometriosis was significantly higher in the ovarian cancer group (OR 1.46). The OR were significantly increased in the hystotypes: Clear cell ovarian carcinoma (CCC) (OR 3.05), Endometrioid ovarian carcinoma (EnOC) (2.04), and Low-grade serous ovarian carcinoma (OR 2.11) [5].

Several studies have reported that atypical endometriosis (AE) should be considered as the direct precursor of CCC and EnOC, as AE has been identified as contiguous with these tumor histotypes [6,7]. AE refers to two different histologic findings: cellular atypia, also known as cytologic atypia, and architectural atypia, commonly referred to as hyperplasia [8]. The “road” to malignant transformation is not well established, although several pathways leading to AE and finally to EAOC have been suggested: oxidative stress, cytokines, genetic mutations, and hyperestrogenic environment may have a role in the carcinogenesis from benign endometriosis (BE) to cancer. Two potential scenarios for ovarian endometrioma leading to EAOC have been proposed. The one involves extracellular hemoglobin, iron, and heme (from the repeated hemorrhages occurring in endometriosis), causing cellular oxidative damage via increased reactive oxygen species (ROS) with subsequent DNA damage and resulting mutations. The second involves the persistent production of antioxidants, which would favor a tumor-potentiating environment. Most CCC and EnOC are included in Type I ovarian tumors, since they develop from benign extraovarian lesions that implant on the ovary and can subsequently undergo malignant transformation (i.e., in endometriosis they arise within benign ovarian endometriotic cysts). Type I ovarian tumors are clinically indolent and usually present with low-grade carcinoma [9,10]. Indeed, AE and EAOC share common molecular/genetic alterations such as somatic ARID-1A [11,12] and Phosphatase and tensin homolog (PTEN) mutations [13], Phosphatidylinositol-4,5-Bisphosphate 3-Kinase Catalytic Subunit Alpha (PIK3CA) mutations [14], hepatocyte nuclear factor (HNF)-1b up-regulation [15], loss of estrogen receptor (ER) and progesterone receptor (PR) [16], and rarely P53 mutations [17]. These mutations exhibit the continuum of tumor progression between benign cystic neoplasms and the corresponding carcinomas, such as EnOC and CCC, often through precursor lesions, such as AE. These and other targets have been proposed for the early detection of endometriosis-related cancers, but the clinical application of these novel biomarkers may be difficult since they all need molecular analysis. To the best of our knowledge, this is the first study that systematically reviews the current literature focused on AE-associated biomarkers to offer a general view of available data.

## 2. Materials and Methods

The study was registered “a priori” and accepted for inclusion in the PROSPERO database (ID CRD42021254634). The systematic review was carried out in accordance with the methods proposed by the Preferred Reporting Item for Systematic Reviews and Meta-analysis (PRISMA) guidelines [18]. We performed an advanced, systematic search of the online medical databases PubMed and Medline using the following keywords: “Endometriosis”, “Atypical”, “Atypical endometriosis”, “Marker”, “Biomarker”, and “Atypia”. To optimize the search output, we used specific tools available in each database, such as Medical Subject Headings (MeSH) terms (PubMed/Medline). EndNote software (available online: https://endnote.com, accessed on 1 July 2021) was used to remove duplicate articles. Only full-length manuscripts written in English up to June 2021 were considered. To overcome the low number of published articles, we also decided to include case reports and series in this systematic review of the literature. We checked all citations found in the title and abstract to establish the eligibility of the source and obtained the full text of eligible articles. A flow diagram of the systematic review is shown in Figure 1 (PRISMA template). We also performed a manual scan of the reference lists of the review articles to identify any additional relevant citations. Three review authors (R.V., M.S., and L.B.) independently assessed the risk of bias for each study using the risk-of-bias tool for case–control studies developed by Clarity group [19] according to the following domains: (i) Can we be confident in the assessment of exposure?; (ii) Can we be confident that cases had developed the outcome of interest and controls had not?; (iii) Were the cases properly selected?; (iv) Were the controls properly selected?; (v) Were cases and controls matched according to important prognostic variables or was statistical adjustment carried out for those variables? We graded each potential source of bias as definitely yes (low risk of bias), probably yes (moderate risk of bias), probably no (serious risk of bias), or definitely no (critical, high risk of bias). We summarized the risk of bias judgments across different studies for each of the domains listed. The risks of bias of the included studies are summarized in Appendix A.

## 3. Results

The search revealed *n* = 40 studies eligible for inclusion in this systematic review. Of these, *n* = 39 were finally included [11,14,15,17,20,21,22,23,24,25,26,27,28,29,30,31,32,33,34,35,36,37,38,39,40,41,42,43,44,45,46,47,48,49,50,51,52,53,54]. A flow diagram of the systematic review is shown in Figure 1 (PRISMA template). The main characteristics of the included studies are summarized in Table 1.

To make the paragraph easier to read, the results derived from the various studies will be presented based on the biological role of the investigated molecules. It is interesting to note that all the above-reported data derive from immunohistochemical analysis, since there are no studies that assess hematological or serum biomarkers for atypical endometriosis.

### 3.1. PI3K/AKT/mTOR Pathway

The phosphatidylinositol 3-kinases (PI3K)/AKT/mTOR pathway is an intracellular signaling pathway that is important in regulating the cell cycle. In many cancers, this pathway is overactive, thus reducing apoptosis and allowing proliferation.

There are many known factors that enhance the PI3K/AKT pathway, including Epidermal growth factor (EGF), Sonic hedgehog (shh), Insulin-like growth factor 1 (IGF-1), and insulin. Both leptin and insulin recruit PI3K signaling for metabolic regulation. The pathway is antagonized by various factors, including PTEN. Moreover, this kinase cascade is highly interconnected with other pathways that regulate cell proliferation, such as the RAS/MAPK/ERK pathway. K-Ras GTPase, which is encoded by the KRAS gene, once activated can upregulate PI3K cell signaling receptors [9].

Five out of 39 studies investigated the correlation between PIK3CA mutations and the presence of AE [14,34,41,42,43]. All of these studies reported concordant mutations in PIK3CA either in cancer or in contiguous AE. PIK3CA is a protein coding gene and, as an integral part of the PI3K pathway, it has long been described as an oncogene. In their retrospective study, Yamamoto et al. [14] found that somatic mutations of the PIK3CA gene were detected in 10/23 (43%) CCC, and in all cases, the type of mutation was H1047R in the kinase domain. The same mutation was retrieved in 6/8 non-atypical endometriotic lesions (75%) and 7/8 atypical endometriotic lesions (88%). These data suggest that mutations of the PIK3CA gene occur in the putative precursor lesions of CCC as very early events in tumorigenesis, probably initiating the malignant transformation of endometriosis in AE, and finally in cancer. The same group of authors showed that somatic mutations of PIK3CA were detected in 17 (40%) tumors, and the majority (71%) of these were ARID1A-deficient carcinomas. All of the 6 PIK3CA-mutation-positive typical endometriosis (TE) were immunohistochemically ARID1A deficient; at the same time, all of the 6 AE harboring PIK3CA mutations were ARID1A deficient [34]. They speculated that the coexistence of the loss of ARID1A expression and PIK3CA mutations suggested a possible cross-talk between the regulation of ARID1A expression and the PI3K signaling pathway during ovarian CCC development.

Confirming their results, Anglesio et al. conducted a whole-genome shotgun sequencing (WGSS) of somatic mutations within 7 cases of CCC. In each specimen, at least one specimen of AE and TE was present for each case. They found that the fraction of detectable somatic mutations that was shared between endometriosis and patient-matched carcinoma ranged from 15 (distant TE) to 98 % (adjacent AE). With the exception of PIK3CA and ARID1A, the constituents of the conserved mutations were generally unique. In cases with somatic ARID1A and/or PIK3CA mutations, they were consistently found to be present across all cancer specimens, as well as any AE or TE from those cases [41]. Matsumoto et al. also found that PIK3CA mutations were observed either in 31.4% of EnOC and 35.7% of CCC, or in five coexisting cases of AE and in 2 cases of non-AE [42]. They also observed that a significantly different expression of Akt, another component of the PI3K pathway, was evident between the two types of EAOC and their coexisting AE and non-AE. Finally, Er et al., after DNA extraction and sequencing of 8 CCC and 2 EnOC, observed that PIK3CA was the most frequent mutation, and it was also observed in concomitant AE in 3/6 cases (50%), even if they reported that only in one case (16%) ARID1A and PIK3CA were both mutated [43]. Only one study assessed the relationship between p-mTOR and the presence of AE [40], showing that the frequency of mTOR immunopositivity was 58% in non-AE, 63% in AE, and 77% in EAOC, with an increasing trend from benign and precursor lesions toward cancer.

Five out of 39 studies reported PTEN mutations associated with AE, with controversial results [28,37,40,45,50]. PTEN acts as a tumor suppressor gene through the action of its phosphatase protein product, which negatively regulates the PI3K/AKT/mTOR pathway.

In their retrospective study, Ali-Fehmi et al. reported that loss of heterozygosity (LOH), microsatellite instability (MSI), and mutations that lead to functional inactivation of PTEN were present either in EAOC, or in endometriosis and AE, suggesting that these genetic alterations may be very early in tumor progression and confirming their precursor status. Moreover, significant differences in LOH were seen between endometriosis (4.3%) and EAOC (23.5%) at D10S608, suggesting that LOH at D105608 may possibly be an important molecular event in the progression of endometriosis to carcinoma [28].

Concomitant PTEN mutations in cancer and coexisting AE were also reported in another retrospective study [37]. Suryawanshi et al. demonstrated that PTEN and K-ras mutations in mice lead to complement upregulation [40]. Gene expression analyses revealed that the complement pathway was most prominently involved in both endometriosis and EAOC. Moreover, from an immune gene expression analysis, they found that AE shared the EAOC pattern. Once again, AE seems to be the precursor of cancer.

In contrast to these findings, Ma [45] and Jiao [50] showed that PTEN expression in cancer was not consistent with its expression in AE. The former in his retrospective study concluded that PTEN and p53 mutation frequency in EAOC were significantly higher compared to AE and endometriosis. The latter, instead, in a case report study, showed a negative immunohistochemical (IHC) PTEN expression both in EAOC and AE. This study is obviously limited by its low population and study design.

Finally, 3 out of 39 studies investigated Kras expression in AE compared to cancer and TE [25,40,48]. Two of them demonstrated that Kras is present either in cancer or in AE and that its expression is related to a higher complement pathway [40] and NFkB signaling activation in AE and EAOC [48]. In contrast to these findings, in a case series study, Amemiya showed that KRAS mutations may be a late event in the malignant transformation to cancer, since they have been observed only in cancer but not in AE or TE [25].

### 3.2. ARID1a and BAF250

BAF250a, the protein encoded by ARID1A, is one of the accessory subunits of the SWI–SNF complex believed to confer specificity in the regulation of gene expression. In analyzing the correlations between mutations or other aberrations in ARID1A and BAF250a expression in ovarian cancer and contiguous AE, eight studies detected possible correlations [11,33,34,37,43,44,46,49], with a rate of ARID1A mutations and BAF250A loss of expression in AE ranging from 23.8 to 100%. In particular, the study conducted by Wiegand et al. found two patients with ovarian CCC carrying ARID1A mutations in contiguous AE; samples of both the ovarian CCC and AE demonstrated loss of BAF250a expression maintaining his expression in the distant typical endometriotic lesion [11].

Similarly, Yamamoto et al. found that 61% of endometriosis-associated ovarian CCC were ARID1A-deficient. Among them, 86% of TE and 100% of AE were judged as ARID1A deficient [34]. The mutational status in 409 cancer-associated genes of 10 Taiwanese patients with EAOC (8 EnOC; 2 ovarian CCC) showed that ARID1A was mutated in 50% of cases. In their concomitant AE, ARID1A was mutated in 33.3% of cases (2/6) [43].

A retrospective study conducted by Xiao et al. examining molecular alterations found loss of BAF250a expression in about 20% of TE, and a similar pattern of expression in AE (38.5%) and CCC (57.7%) in comparison to papillary serous carcinoma [33].

Lai et al., when comparing the expression of BAF250a among 79 cases of EAOC (40 CCC; 33 EnOC; 4 serous carcinomas; 1 adenosquamous carcinoma, and 1 adenosarcoma), revealed loss of BAF250a in 37 (47%) cases but without statistical significance among different subtypes. Interestingly, all staining results were similar between EAOC and contiguous AE [37].

Kato et al. observed BAF250a deficient expression in 30% (29/97) of ovarian CCC, and among them, they found more frequently the concurrence of endometriosis compared with BAF250a retained cases (*p* < 0.05). The frequencies of BAF250a deficient expression were 19% (6/31) in TE, 26% (10/38) in AE, 39% (15/38) in synchronous with endometriosis CCC, and 6% (1/18) in solitary endometriosis [44].

A retrospective study conducted by Stamp et al. showed loss of BAF250a expression in 14/35 cancer patients. Among them, AE was present in 10 cases showing BAF250a loss in 60% of cases. None of the 8 cases of AE not associated with cancer had BAF250a loss [46].

A Spanish prospective study conducted by Niguez-Sevilla et al. analyzing 185 patients with endometriosis reported AE in 23 cases (12.43%), of which 14 (60.86%) had endometriosis alone and the other 9 (39.13%) had EAOC. They found a higher loss of BAF250a expression in the AE versus TE group (23.8% vs. 3%) (*p* = 0.004) [49].

### 3.3. Estrogen Receptors and Progesterone Receptors

Since estrogen and progesterone receptors are expressed in endometriotic tissue, the loss of their expression in EAOC may be part of carcinogenesis and indicative of cell dedifferentiation. Ten studies assessed the expression of hormonal receptors in AE, with controversial results [24,27,33,37,39,47,50,51,52,53]. Five showed a significant ER and/or PR loss of expression in AE and EAOC.

A reduction in the expression of hormone receptors from endometriosis to AE has been described by Lenz et al., suggesting their role in diagnosing AE. The authors found a decrease in ER (on average 56%) and PR (less than 1%) expression in atypical ovarian endometriosis compared to lymph node endometriosis and deep infiltrating endometriosis [53].

The analysis conducted by Xiao et al. detected the expression of ER in only 7.7% (2/26) of ovarian CCC and 91.8% (22/24) of papillary serous carcinoma cases, while PR expression was not significantly different between them. Fifty percent of CCC (13/26) showed concomitant AE with a loss of ER and PR expression. In particular, both AE and adjacent CCC had a similar profile, with a loss of ER (84.6% vs. 92.3%) and PR (76.9% vs. 84.6%) expression. The concurrent rate of loss of BAF250a expression, HNF-1b upregulation, and loss of ER expression was not observed in any TE, was increased to 23.1% in AE, and was further increased to 42.3% in CCC [33].

The retrospective study conducted by Lin et al. in 12 patients with CCC showed how the positive ratio of ERb expression gradually reduced from ovarian endometriosis (83.3%) to AE (33.3%) to cancer (0%) with minimal changes of ERa expression during the process. Conversely, the poorest intense expression of steroid receptor RNA activator protein (SRAP) was detected in the cells of endometriosis, gradually increasing in the process of malignant transformation, with the most intense expression in cancer [39].

Concordant with these findings, the study conducted by Andersen et al. on 19 EAOC (5 CCC; 14 EnOC) found decreased ERα and PR protein levels from benign endometriosis to EAOC while ERβ expression increased incrementally from benign endometriosis to EAOC [47].

A case report of a poorly differentiated mucinous carcinoma with signet ring cells and concurrent endometriotic cyst with atypical features showed that the tumor cells were negative for estrogen and progesterone receptors upon immunohistochemistry [50].

On the contrary, as demonstrated by other studies, estrogens appear to be a mitogen for endometriosis, and estrogen-receptor positivity is observed in EnOC. However, CCC does not present estrogen receptor positive expression in tumor tissue. Del Carmen et al. found that 23% of EAOC and 100% of AE tissue blocks, respectively, exhibited positive staining for ER (*p* < 0.05), while 35% of EAOC and 100% of AE samples showed positive staining for PR (*p* < 0.05) [24]. A case series evaluating the expression of steroid hormone receptors in the pathological progression from endometriosis to AE to CCC showed a gradual reduction in ERa and PRA expression. In the oncogenesis of EnOC from TE to AE and finally to cancer, a gradual increase in ER alpha was observed [27]. Lai et al. evaluated 79 cases of EAOC (40 CCC; 33 EnOC; four serous carcinomas; one adenosquamous carcinoma; one adenosarcoma). The analysis resulted in positive staining for ER more often in EnOC (30/33, 91%) than in CCC (3/40, 8%) and serous carcinomas (0/4, 0%) supporting the suggestion that estrogen-dependent ovarian cancer arising from endometriosis is more associated with EnOC than CCC. All staining results were similar between AE and contiguous EAOC [37]. Finally, two Romanian studies reported different results concerning hormone receptor expression in AE and EAOC [51,52]. In fact, Păvăleanu et al. showed negative ER immunoexpression in 29.03% cases of endometriotic areas, while PR revealed a negative score in 38.70% of cases. The authors also identified a particular immunoexpression pattern of ER and PR in the stromal cells (32.25% of cases had positive ER immunoreaction, while 35.48% of cases were PR positive). With regards to EAOC, ER immunoexpression had a negative score of 21.05% of cases, while PR immunoexpression revealed a negative score in 47.36% of cases, with a stromal expression for both ER and PR significantly higher in EAOC when compared to endometriosis [51]. Higher values of ER expression in endometriosis associated with high-grade serous ovarian cancer than in TE/AE associated with EnOC were found by Penciu et al. Moreover, higher values of ER expression were also recorded in ovarian cancer than in endometriotic foci [52].

### 3.4. Transcriptional and Nuclear Factors

Seventeen out of 39 studies [15,17,21,22,23,24,26,27,31,32,33,37,42,45,50,52,53] have investigated the expression of some transcriptional and nuclear factors involved in cancer development, aiming to point out any similarities between AE and ovarian tumors.

Among them, ten studies (six retrospective studies, two case series, one case report, and one comparative study) have analyzed the expression of p53, a tumor suppressor gene. However, the results were not concordant. Lenz et al. [53] found higher and stronger p53 alterations in AE compared to benign ovarian endometriosis. Nezhat et al. observed positive p53 staining in cancers and in benign ovarian endometriosis next to the tumor [23]. Similar results were obtained from the retrospective study of Saìnz de la Cuesta, where more than 50% of cancer and AE had p53 overexpression, with an increasing trend going from benign endometriosis to cancer [26]. The same findings were reported in the case series of Akahane et al., the case report of Jiao et al., and the retrospective study of Ma et al. [27,45,50]. In contrast, Bayramoglu et al. found p53 alterations in neither AE nor in 17 out of 20 cancer tissues analyzed [22]. In another retrospective study, only 13% of cancers stained positive for p53. All staining results were similar between AE and contiguous EAOC [37]. The same results were obtained from Akahane et al. [17] since only 8/26 cancer tissue analyzed had p53 mutations. Moreover, no p53 alterations were found neither in endometriosis nor in AE tissue coexisting with cancer. Penciu et al. [52] observed p53 alterations only in two cases of ovarian serous carcinoma. They did not find the same results in EnOC and corresponding AE. Seven studies analyzed ki-67 which represents an excellent marker to determine the growth fraction of a given cell population, particularly used to assess cancer growth. Akahane, Ogawa, and Yamamoto found a mean Ki67 labeling index increased from endometriosis through AE to cancer [17,21,31], while Del Carmen, Penciu, and Lenz did not observe any significant findings [24,52,53]. Another tumor suppressor protein that has been investigated is p16, a protein that slows cell division by slowing the progression of the cell cycle from the G1 phase to the S phase. Tumor cells were positive for p16 only in the case report of Yurong Jiao [50] and in two cases of ovarian serous cancer in the case series of Penciu [52]. In both studies, AE had a different pathway from cancer, not exhibiting p16 at immunohistochemical analysis. Among the transcription factors, hepatocyte nuclear factor-1 beta (HNF1-B), a protein that binds to specific regions of DNA and regulates the activity of other genes, has been analyzed in 3 out of 39 studies, with concordant HNF-1b expression between AE and EAOC. In particular, 30% of cancer cases showed positive staining for HNF-1 beta, with similar staining results in adjacent AE [37]. In another study, all CCC cases demonstrated strong nuclear immunostaining for HNF-1Beta, while benign endometriotic cysts were negative. 4 cases of AE showed nuclear staining for HNF-1beta in the atypical endometriotic epithelium [15]. In another study, it was found that AE and CCC had a similar profile with HNF-1beta up-regulation (53.8% vs. 92.3%) [33]. Another transcription factor investigated is hypoxia-inducible factors (HIFs), which respond to a decrease in available oxygen in the cellular environment. Kato and colleagues found an increasing trend in HIF immunopositivity from TE (5%) to AE (37%) and then to cancer (95%) [32]. However, HIF expression could differ according to cancer histotype: significant differences in the expression of HIF-1α were evident between the two types of EAOC (CCC and EnOC) and their coexisting AE and non-AE [42].

### 3.5. Growth Factors and Their Receptors

Among the growth factor receptors, platelet-derived growth factor-A (PDGF) and its receptor-a/b (PDGFR-a/b) showed increased positivity in accordance with higher cytologic atypia in the putative precursors: 71, 47, and 59% in the 17 TE, 84, 73, and 84% in the 19 AE and 97, 97, and 100% in the 31 CCC. Positivity for PDGF-B decreased in accordance with increased atypia in endometriosis coexisting with CCC: 35% in TE, 11% in AE, and 5% in coexisting carcinomas [38]. In addition, the vascular endothelial growth factor (VEGF) stained positively in 94.1% EAOC compared with only 12.5% of the AE [32]. Akahane et al. [35] found an increased Receptor tyrosine-protein kinase 2 expression (c-erbB-2) in CCC and EnOC, first observed in AE, but Saìnz de la Cuesta et al. [34] did not observe any differences in (erbB-2) between cancer and AE because it was never expressed.

### 3.6. Others

Other molecular markers have been investigated in a solitary or a few studies. The results from these studies can be reviewed in Table 1. Among these, the most frequently assessed was Bcl-2 (B-cell lymphoma 2). It is encoded in humans by the BCL2 gene and is the founding member of the Bcl-2 family of regulator proteins that regulate cell death (apoptosis) by either inhibiting (anti-apoptotic) or inducing (pro-apoptotic) apoptosis. Three out of 39 studies assessed the correlation between Bcl-2 immunohistochemical expression and AE [23,26,51]. Two of them found that a higher Bcl-2 rate in endometriosis had a statistically significant association with more aggressive tumor behavior, starting from TE toward AE and finally cancers (both CCC and EnOC histotypes) [23,51]. On the contrary, Sainz de la Cuesta and coworkers did not find any significant difference in Bcl-2 expression either in EAOC or in AE compared to TE [26].

## 4. Discussion

Atypical endometriosis has been observed in 12–35% of ovarian endometriosis, and it is estimated that around 60% to 80% of all EAOC occur in the presence of AE, often found in direct continuity with the tumor [55]. Evidence suggests that AE could be a transitioning entity from benign lesions to malignant variants. The pre-malignant, “atypical” lesions, are defined by several histologic criteria, including large nuclei with moderate to marked pleomorphism, increased nuclear-to-cytoplasmic ratio, cellular crowding, stratification, or tufting [55,56]. It has been reported that AE actually refers to two different histologic findings: cellular atypia, also known as cytologic atypia, and architectural atypia, commonly referred to as hyperplasia. The diagnosis of endometriosis with architectural atypia is important because patients with hyperplasia-type endometriosis may be at a higher risk of developing EAOC [49]. This variability in the incidence of the disease may be due to a difficult histological diagnosis, which is still far from being standardized among medical centers worldwide. Moreover, the new conceptualization of the histological pattern, and the differences that emerged in prognostic significance between cytological AE and hyperplastic AE should strongly encourage a revision of this classification in order to better understand which type of AE is actually to define a “high-risk” disease. Reflections can be made on the meaning of a “preneoplastic lesion”: determining whether the mere presence of endometrium at ectopic sites should be considered “per se” a premalignant condition seems crucial and constitutes the conceptual base of any strategy aimed at reducing EAOC mortality. Lesions are defined as “precancerous” based on definite epidemiologic, morphologic, molecular, and biologic criteria that imply the acquisition of genetic, karyotypic, structural, or functional changes in a cluster of cells that differentiate them from the surrounding normal tissue [57]. In other words, premalignant lesions such as AE should reflect an intermediate stage along the pathway leading to cancer. When enough genetic changes have occurred, modifications in appearance and function are observed but not yet associated with typical malignant behavior. In recent years, the literature has focused on the relationship between endometriosis and ovarian cancer. Indeed, sequencing and immunohistochemical studies have revealed that mutations found in endometriosis-associated cancers are found in adjacent endometriosis, supporting the theory of a clonal relationship between benign and malignant counterparts and confirming that the cancers have arisen from the endometriotic lesions probably through an “intermediate” premalignant step. In addition, gene encoding b-catenin (CTNNB1) mutations in 16–53.3%, PTEN mutations in 14–20%, and ARID1A mutations in 30–55% of cases were found in EnOC. PIK3CA mutations in 20–40% and ARID1A mutations in 46–57% of cases are found in ovarian CCC [58].

This systematic review has been conducted with the aim of identifying all biomarkers associated with AE in order to have a “pre-histological” diagnosis of this premalignant entity. Unfortunately, in regards to the results of our systematic review, none of the above discussed biomarkers has such strong evidence that could justify their systematic clinical use in the management of endometriosis and AE. The most frequently detected mutations in AE are ARID1A, genes involved in PI3K pathway (i.e., PIK3CA), genes encoding for ER and PR, KRAS, and PTEN. Interestingly, somatic driven mutations in KRAS, PTEN, PIK3CA and ARID1A have been also observed in more than 26% of cases of deep infiltrating endometriosis lesions, which are associated with virtually no risk of malignant transformation [59]. Therefore, a specific role seems to be played by the ovarian microenvironment in increasing the risk of malignant derailment [60]. Karnezis concurs that the ovarian microenvironment seems to be essential for the malignant transformation of endometriosis because many endometriotic lesions are located outside the ovary, including the pelvic peritoneum, but carcinomas at such sites are rare [61]. Indeed, it was proposed for the endometrioma’s neoplastic transformation as a hypothetical model called the “two-hits” hypothesis [9]. Reactive oxygen species due to free heme and catalytic iron contained in the trapped blood in endometriomas may lead to increased oxidative stress and DNA damage in the epithelial layer of endometriomas. This may result in mutations and epigenetic changes, including mutations in the tumor suppressor gene ARID1A leading to AE. Possible second-hit mutations, as well as activation of the PI3K-AKT-mTOR pathway allow the mutated cell to escape apoptosis caused by increased oxidative stress. The accumulation of oncogenic mutations in AE may ultimately lead to the development of endometriosis-associated ovarian CCC and endometrioid carcinomas. ARID1A loss and activation of PI3K/AKT functionally cooperate in ovarian carcinogenesis and suggest that ARID1A-loss occurs early and that it may be “addicted” to PI3K/AKT oncogenic signaling [9]. In line with this hypothesis and with the reduction of the hormone dependency of CCC, oxidative stress has been shown to act as a physiological regulator of estrogen receptors. Contrary to CCC, EnOC is generally estrogen sensitive and associated with hormonal risk factors. ERa has been shown to represent an independent prognostic marker for EnOC, while nuclear ERa is barely detectable in CCC [62]. Inactivating ARID1A mutations are the most common molecular genetic alteration reported thus far in CCC and EnOC, but a higher frequency of ARID1A mutations has been detected in CCC (46–57%) compared with EnOC (30%). The fact that no differences in clinical behavior were observed comparing BAF250a-positive versus BAF250a-negative cancers may be the basis for supporting the idea of a marker for genomic instability without driving the phenotype: BAF250a appears to be an early event in most of these cases [46,63]. Based on results from the systematic review, there is a remarkable association between BAF250a loss in cases of AE and TE contiguous to BAF250a-deficient EAOC with a higher frequency of inactivating ARID1A mutations detected in CCC compared to EnOC [11,33,34,37,43,44,46,49]. Concordantly with these findings, also PI3K/Akt/mTOR molecular pathway seems to be altered in AE and in EAOC in all of the studies included, with a similar pattern between the two diseases [25,28,31,32,34,37,40,42,43,45,48,50]. Since ovarian cancer is considered a hormone-responsive cancer, its correlation to PR and ER immunoexpression has a major importance in clinic-pathological manifestations of ovarian carcinoma, including those associated with endometriosis [64,65]. Actions of estradiol are mediated by two isoforms of ERs (ERa and ERb) that differ in their tissue distributions, their ligand-binding specificity, and affinity: ERa has been shown to represent an independent prognostic marker for EnOC, while nuclear ERa was poorly detectable in CCC [62,66]. Since progesterone is modulated by the expression of both isoforms of the specific receptor (PR-A and PR-B), it is involved in the pathogenesis of endometriosis and EAOC [65]. Thus, from the results of the systematic review, it seems that alteration of steroid receptor immune expression is correlated to ovarian endometriosis and endometriosis-related carcinogenesis in a hormone-dependent manner with regards to EnOC and in a hormone-independent way concerning CCC.

Associated AE lesions seem to have the same biological expression as their adjacent Cancer histotypes [15,67,68,69,70,71,72,73,74,75]. Moreover, ER and PR expression seems to be higher in AE as compared to EAOC, and lower when compared to endometriosis. The gradual loss of ER and PR expression from endometriosis to EAOC carcinogenesis suggests that hormone receptor staining may be proposed as a marker for premalignant or malignant lesions in endometriosis. A reduced HNF 1-beta expression in AE as compared to EAOC, and in particular to CCC, and a higher HNF-1 beta expression in AE as compared to TE has also been reported in our systematic review [15,33,37]. The main limit to the use of these molecular markers proposed for the early detection of endometriosis-preneoplastic lesions is that their clinical application may be difficult since they all need expensive molecular analyses. Moreover, strong evidence supporting their systematic use in clinical practice is still lacking. The presence of a serum non-invasive marker for the presence of AE could be more effective and easy to use. Inflammatory parameters, such as the neutrophil-to-lymphocyte ratio (NLR) and platelet-to-lymphocyte ratio (PLR), have been found altered in the peripheral blood in patients with endometriosis, thus suggesting that systemic perturbations may contribute to the pathogenetic process of the disease [76]. NLR and PLR have also gained more and more space in the diagnostic and prognostic management of ovarian carcinoma [77]. Indeed, systemic inflammation contributes to cancer initiation and progression by promoting cell proliferation, angiogenesis, and gene repair [78]. To the best of our knowledge, there are no studies in the literature evaluating the possible role of NLR and PLR in predicting AE and its risk of malignant transformation. This is even more important if we think that, according to the literature, women with endometriosis had a tripled and doubled risk for CCC and EnOC subtypes, and a more frequent localized form of the disease when cancer arises in endometriosis [79]. Even if cancer arising in endometriosis seems to be characterized by a better prognosis, the early detection of preneoplastic lesions could really impact the quality of life of women with endometriosis. Our study has some limitations: (1) The non-homogeneous definition of AE in the studies included in the systematic review; (2) Heterogeneity between studies (i.e., study design) included in the systematic review; and (3) Low number of studies included in the systematic review. These limitations suggest that the results should be interpreted with caution until validated by future research projects providing more detailed, well-designed, and standardized data collection.

## 5. Conclusions

Atypical endometriosis can be a transitioning entity from endometriosis to endometriosis-associated ovarian cancers. In our systematic review, we found 39 studies assessing numerous molecular targets of AE, such as immunohistochemical expression of BAF250, PIK3CA, PTEN, HNF-1beta, ER, and PR. Unfortunately, these molecular biomarkers of AE require expensive molecular analysis, histological examination is always needed, and none of them has such strong evidence to justify their systematic use in the management of the neoplastic risk of endometriosis. Further studies are needed to validate evidence on available biomarkers for the presence of AE, which is a high oncologic risk condition. Moreover, the introduction of novel serum biomarkers could be useful for the non-invasive diagnosis of AE.

## Figures and Tables

**Figure 1 ijms-23-04425-f001:**
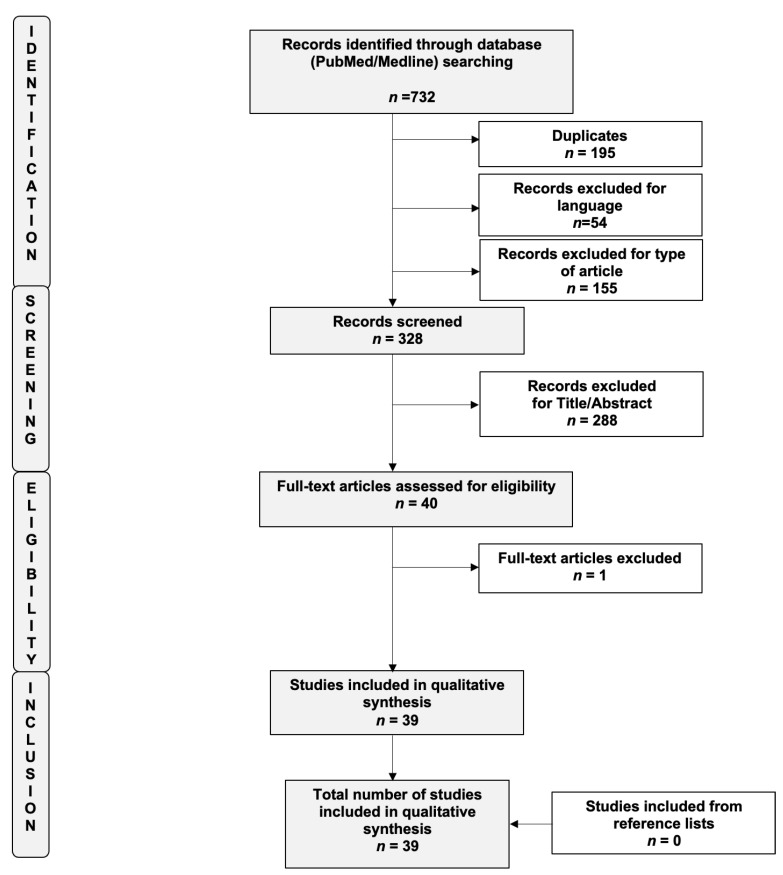
Flow diagram of the search strategy, screening, eligibility, and inclusion criteria (PRISMA template).

**Table 1 ijms-23-04425-t001:** Studies investigating molecular targets associated with atypical endometriosis.

Authors	Date	Type of Study	Immunohistochemical Analysis	Study Period	AE (n), Mean Age ± ds (yo)	EAOC (n), Mean Age ± ds (yo)	Endometrios (n), Mean Age ± ds (yo)	Results
Chalas et al. [20]	1990	Retrospective Study	Average AgNORs * per nucleus	NR	10, NR	10, NR	10, NR	- The mean AgNOR count per nucleus was significantly greater in cancer than in TE and AE. - The mean AgNORs count per nucleus was significantly greater in AE than in TE.
Ogawa et al. [21]	2000	Retrospective Study	Ki-67	1980–1995	29, NR	30 CCC, 3 EnOC, 4 OSC, 51.4 yo (range, 22–80 years)	33, NR	- Ki-67 indices were as follows: ovarian carcinoma, 23.1 + 3.29; AE, 9.9 + 1.73; TE, 2.7 + 0.90. - Significant differences were observed between carcinoma and AE, between AE and TE, and between carcinoma and TE. *
Bayramoglu et al. [22]	2001	Retrospective Study	p53	NR	7, 34.1 ± NR	10, NR	137, 36.1 ± NR	- 2/10 EAOC showed diffuse-strong and 1/10 showed focal-strong p53 tumor suppressor gene expression. - No alterations in the AE group.
Nezhat et al. [23]	2002	Comparative Study	bcl-2, p53	NR	--	24 EnOC, 19 CCC, 40 OSC; NR	30; NR	- bcl-2 was reported to stain 23% of benign endometriotic cysts, 67% of EnOC, 73% of CCC, and 50% of OSC. - 42% of benign endometriotic lesions adjacent to the EnOC and 73% adjacent to CCC were found to stain for bcl-2. - p53 staining was negative in the benign endometriotic cyst group and was positive in 37–55% of the group with tumors. - p53 staining was positive in 25% of the benign endometriotic lesions next to the EnOC and in 9% of the benign endometriotic lesions next to CCC.
Del Carmen et al. [24]	2003	Retrospective study	VEGF, Ki-67, ER, PR	NR	17, NR	8, NR	--	- 94.1% EAOC stained positively for VEGF, compared with only 12.5% of the AE. - 23.5% EAOC stained positively for ER compared with 100% AE. - 35% EAOC stained positively for PR compared with 100% AE.
Amemiya et al. [25]	2004	Case series	K-ras mutation MSI	1987–1999	--	12 EnOC, 56 ± 7.3 (range 43–68)	--	- 5/12 had MSI (3/5 high MSI). - 1/12 K-ras codon 12 mutation.
Sáinz de la Cuesta et al. [26]	2004	Retrospective study	p53, c-erbB-2, MIB1, and Bcl-2	1948–1999	6 45.5 yo (range 33–78)	17 45.5 yo (range 33–78)	17 45.5 yo (range 33–78)	- 14 out of 17 (82.4%) EAOC associated with endometriosis and all AE had p53 overexpression. Only 2 of 17 (11.8%) endometriosis patients had mutant p53. - Increased expression of MIB1 (0.073) in the cancer and AE groups, and no differences in expression of Bcl-2 or c-erb-B-2 (never expressed). - The sensitivity and specificity of p53 as a marker for AE/EAOCs were 87%; CI 95% (73.2–100%) and 92% (80.6–100%). - p53 and MIB1 overexpression was lowest in the endometriosis and highest in the EAOC, with an intermediate expression in the AE.
Akahane et al. [27]	2005	Case Series	ERα, ERβ, PRA, PRB p53, Ki-67, c-erbB-2, EGFR	1993–2000	--	4 CCC, 56,5 ± 1.11 4 EnOC, 56,8 ± 7.8	--	**Both CCC and EnOC:** - No changes in ERβ and PRB; **CCC:** - Gradual reductions of ERa in 3/4 CCCs, and of PRA in 2/4 CCCs (in particular was first noted in areas of AE); - Increased c-erbB-2 expression (first observed at sites of AE); - Disappearance of steroid hormone dependency might be involved in the malignant transformation of endometriosis into CCCs. **EnOC** - Gradual increases in ERa expression from endometriosis to AE to carcinoma; - Increased p53 and Ki-67 expression (first observed at AE sites).
Kato N et al. [15]	2006	Retrospective Study	HNF-1beta	NR	4, 32–79	30 (17 CCC), 32–79	40, NR	- All of the 30 CCCs showed strong nuclear immunostaining for HNF-1beta. - No significant difference between the HNF-1beta score for CCC with endometriosis (17 cases) and that for CCC without endometriosis (13 cases). - Among 12 CCC in which endometriotic epithelium was identified, 9 cases showed distinct nuclear staining for HNF-1beta in the endometriotic epithelium. The epithelium showing HNF-1beta expression was that of AE in 4 cases (13%), while it was that of endometriosis showing inflammatory atypia in 5 cases. - Most of the epithelium of endometriotic cysts was negative for nuclear staining for HNF-1beta.
Ali-Fehmi et al. [28]	2006	Retrospective Study	LOH and MSI at PTEN (10q23.3) in loci D10S215, D10S608, and D10S541	NR	12, NR	20, NR 5 CCC 7 EnOC 8 OSC	23, NR	- LOH was present in endometriosis (30%), AE (25%) and ovarian carcinoma (40%). - A high frequency of MSI was found in endometriosis (82.6%) and AE (75%) compared to EAOC (53%). - Significant differences in LOH were seen between endometriosis (4.3%) and ovarian carcinoma (23.5%) at D10S608.
Akahane et al. [17]	2007	Retrospective Study	p53	1993–2002	--	13 CCC, 55 ± NR (range 29–62) 9 EnOC, 51 ± NR (range 42–76)	7, 34 ± NR (range 30–43)	- No p53 mutations in solitary endometriosis or endometriosis coexisting with EnOC. - CCC cases, p53 mutation was observed in 4/13 endometriosis cases and in 4/13 cancer cell cases. - In 1/9 of EnOC, the cancer tissue had a nonsense mutation of the p53 gene.
Finas et al. [29]	2008	Retrospective Study	L1CAM	NR	14, NR	--	17, NR	- 2/14 were L1 positive. - 13/17 were L1 positive.
Yamamoto et al. [30]	2008	Retrospective Study	PDGFR-a, PDGFR-b, PDGF-AB	1987–2005	19, NR	31 CCC, NR	17, NR	- Positivity for PDGFR-a/b, PDGF-A increased in accordance with higher cytologic atypia in the putative precursors: 71, 47, and 59% in the 17 TE, 84, 73, and 84% in the AE and 97, 97, and 100% in the 31 CCC. - Positivity for PDGF-B decreased in accordance with increased atypia in endometriosis coexisting with CCC: 35% in TE, 11% in AE, and 5% in coexisting carcinomas.
Wiegand et al. [11]	2010	Case reports of 2 patients	BAF250a	NR	2 AE contiguous to CCC	2 CCC	2 distant endometriosis to CCC	- ARID1A mutations and loss of BAF250a expression evident in CCC and contiguous AE but not in distant endometriotic lesions. - Both regions of endometriosis differ from the CCC in their lack of HNF-1β expression (with weak expression in the contiguous AE).
Yamamoto et al. [31]	2010	Retrospective Study	p27Kip1, Skp2, Cks1, cyclin A, and cyclin E, and Ki67 labeling index	1988–2007	15, NR	23 CCC, NR	31, NR	- The cell-cycle regulators examined were overexpressed (Skp2, Cks1, cyclin A and cyclin E) or downregulated (p27Kip1) significantly more frequently in the CCC than in the adjacent endometriosis *. - The frequency of Skp2 overexpression was significantly higher in AE than in endometriosis, and the frequency of Skp2 and cyclin A overexpression was significantly higher in CCC than in AE *. - The mean Ki67 labeling index increased from endometriosis (8.4%) through AE (21.4%) to CCC (46.9%) *.
Yamamoto et al. [14]	2011	Retrospective Study	PIK3CA	1986–2007	8, NR	23 CCC, NR	8, NR	- Direct sequencing identified mutations in 10 (43%) of the 23 endometriosis-associated CCC (H1047R substitution). - Six (75%) of eight non-atypical endometriotic lesions and seven (88%) of eight atypical endometriotic lesions contained the H1047R mutation.
Kato et al. [32]	2012	Retrospective Study	p-mTOR, HIF-1, Glut1	1987–2005	16, NR	36 CCC, NR	21, NR	- The frequencies of immunopositivity for p-mTOR, HIF-1 a, and Glut1iwere 58, 5 and 16% in the non-AE; 63, 37 and 50% in AE; 77, 95 and 95% in the EAOC. - p-mTOR, HIF-1a and Glut1 were positive in 10, 5, and 19% of the solitary endometriosis, respectively.
Xiao et al. [33]	2012	Retrospective study	BAF250a, HNF-1b, ER and PR	1995–2010	13	26 CCC	36	- both AE and adjacent CCC had a similar prifile, with a loss of BAF250a expression (38.5% vs. 57.7%), HNF-1b up-regulation (53.8% vs. 92.3%), and loss of ER (84.6% vs. 92.3%) and PR (76.9% vs. 84.6%) expression. - About 20% of BE had loss of BAF250a expression, 33% with HNF-1b up-regulation, 23% loss of ER expression, and 50% loss of PR expression, respectively. - The concurrent rate of loss of BAF250a expression, HNF-1b up-regulation, and loss of ER expression was not observed in any BE, was increased to 23.1% in AE, and was further increased to 42.3% in CCC.
Yamamoto et al. [34]	2012	Retrospective Study	ARID1A, PIK3CA	1986–2007	22, NR	28 CCC, NR	22, NR	- ARID1A immunoreactivity was deficient in 17 (61%) of the 28 endometriosis-associated carcinomas. - Among the precursor lesions adjacent to the 23 ARID1A-deficient carcinomas, 86% of the TE (12 of 14) and 100% of the AE (14 of 14), benign (3 of 3), and borderline (6 of 6) clear-cell adenofibroma components were found to be ARID1A deficient. - All 22 solitary endometrioses and 10 endometrioses distant from ARID1A-deficient carcinomas showed diffuse immunoreactivity for ARID1A. - Somatic mutations of PIK3CA were detected in 17 (40%) tumors, and the majority (71%) of these were ARID1A-deficient carcinomas (NS). - All six of the PIK3CA-mutation-positive non- AE were immunohistochemically ARID1A deficient. All of the 6 AE harboring PIK3CA mutations were ARID1A deficient.
Yamamoto et al. [35]	2012	Retrospective Study	MET	1987–2006	10, NR	5 CCC, NR	10, NR	- All the 10 non-AE examined were found to exhibit no gain of MET by double in situ hybridization assay, and all showed weak immunoreactions for MET. - Of the 10 AE, 1 (10%), 4 (40%), and 5 (50%) lesions were defined as exhibiting no gain, low-level gain, and high-level gain of MET, respectively. - Of the 5 CCC wherein adjacent AE harbored a high-level gain of MET, all the corresponding carcinoma components examined also showed high-level gain of MET and overexpression of MET.
Yamamoto et al. [36]	2012	Retrospective study	ACTN4 gene (encoding for Actinin-4)	1986–2007	12 adjacent to tumor, of which:-9 both AE and BE -3 only AE	16 adjacent to tumor, of -9 both AE and BE which:-7 only BE	19 CCC	- All 16 BE showed no gain of ACTN4 or actinin-4 overexpression; - 50% of the AE and 75% of the borderline CCAFs showed low-level gains of ACTN4 and actinin-4 overexpression, respectively.
Lai et al. [37]	2013	Retrospective Study	ER, HNF-1 beta, p53, PTEN, BAF250a, COX-2	2001–2011	--	79 ((33 EnOC; 40 CCC; 4 OSC; 1 ASC; 1 AS), NR	--	- Positive staining for ER, HNF1ß, p53, and COX-2 were identified in 34 (43%), 30 (38%), 10 (13%), and 44 (56%) cases. - Loss of PTEN and BAF250a were noted in 29 (37%) and 37 (47%) cases. - All staining results were similar between AE and contiguous EAOC. - The expression of ER was reversely correlated with that of HNF1ß and correlated with p53 *.
Vercellini et al. [38]	2013	Retrospective study	IMP3	2004-2009	9 (35.1 ± 8.1)	NR	508	- 8 of 9 (88%) AE showed IMP3 expression, no expression in contiguous endometrial benign cells and in benign cyst - Test specificity and PPV: 100%; Sensitivity 88.9%, NPV 99.8%.
Lin et al. [39]	2014	Retrospective Study	SRAP, ERs	2003–2012	12, 32–62 yo	12 CCC, 32–62 yo	24, NR	- The positive ratio of ER- expression inthe patients with CCC gradually reduced from ovarian endometriosis (83.3%) to AE (33.3%) to CCC (0%). - The poorest intense expression of SRAP was detected in the cells of endometriosis showing the most intense expression in the CCC.
Suryawanshi et al. [40]	2014	Retrospective Study	Immune gene expression analysis	NR	15 (48 ± 6.5)	28 (54.8 yo ± 11.6)	NE: 32 (46.5 yo ± 6) BE:30 (40 yo ± 10)	- Immune gene expression analysis revealed different disease categories: controls, endometriosis and EAOC (AE shared an EAOC pattern). - 33% of the patients with endometriosis revealed a tumor-like inflammation profile. - Gene expression analyses revealed the complement pathway as most prominently involved in both endometriosis, AE and EAOC (C3, C4a, C7, CFD, CFB, CFH, MASP1). - Complement proteins were also highly expressed in endometriosis, AE and EAOC epithelial cells at IHC. - Conditional activation of tumor-driving pathways (K-ras activating mutations and PTEN deletion) leads to complement gene upregulation. - C7 knockdown in a mouse model inhibited ovarian cell proliferation.
Anglesio et al. [41]	2015	Whole-genome shotgun sequencing (WGSS)	Overall pattern of somatic mutations within EAOC (CCC and EnOC)	NR	-	7 CCC	7	- DNA copy gains in HNF1B (5/7), PPM1D (5/7), ERBB2 (4/7), STAT3 (4/7); 3q PIK3CA (4/7), ARID1A (4/7), including one high-level gain of PIK3CA observed in case 7. - No amplification of MET or amplification of ERBB2.
Matsumoto et al. [42]	2015	Retrospective Study	HIF-1a, iNOS, PIK3CA, pAkt, p65, and HNF-1β, Mutations of the β-Catenin and PIK3CA Genes	2000–2014	--	28 CCC, 35 EnOC, 54.1 (range 22–28)	--	- Mutations in exon 3 of the β-catenin gene were identified in 21 (60%) of 35 EnOCs and in the coexisting non-AE (52.4%) and AE (73.3%) but not in any of the CCC and their coexisting endometriosis. - PIK3CA mutations were observed in 11 (31.4%) of 35 EnOC and 10 (35.7%) of 28 OCCC. The same mutations were detected in coexisting AE and non-AE in 3/11 EnOC and 4/11 CCC. - Significant differences in the expression of pAkt, HNF- 1β,HIF-1α, p65, and iNOS were evident between the two types of tumors and their coexisting AE and non-AE.
Er et al. [43]	2016	Case series	DNA extraction and sequencing	2006–2012	--	8 CCC 2 EnOC	--	- The most frequently mutated genes in cancer were: PIK3CA (60 %; 6/10), ARID1A (50%; 5/10), ETS1, MLH1 and PRKDC (30%; 3/10), and SYNE1 (20%; 2/10). - In concomitant AE: ARID1A (33.3%; 2/6), PIK3CA (50%; 3/6). - 108 mutations in case 1 and 50 mutations in case 2, including MLH1, MSH2, and MSH6 genes involved in the DNA mismatch repair (MMR) system.
Kato et al. [44]	2016	Retrospective Study	BAF250a	1984–2007	38, NR	38 CCC, NR	18, NR	- Concurrence of endometriosis observed more frequently in BAF250a-deficient cases compared with in BAF250a-retained cases *. - BAF250a-deficient expression were 26% (10/38) in AE. - In solitary endometriosis, loss of BAF250a expression was detected in 6% (1/18) of the cases. - A significant difference of BAF250a-deficient expression was observed between EM-related CCCs and CCAF-related CCCs *.
Ma et al. [45]	2016	Retrospective Study	PTEN and p53		10, NR	23, NR	20, NR	- PTEN and p53 mutation frequency in EAOC were significantly higher than that in AE and endometriosis. - There was a significant difference to compare EAOC with endometriosis, and converse to compare with AE, respectively *. - In 2 cases of histological malignant progression, both p53 and PTEN mutations were present. - A decreasing trend of PTEN mutation and an increasing trend of p53 mutation were represented with the increased age, decreased clinical stage, pathological grade, and metastasis.
Stamp et al. [46]	2016	Retrospective study	BAF250a	2005–2008	23 associated with cancer 8 not associated with cancer 45 yo (33–59)	21 EnOC, 50 yo (30-70) 11 CCC, 57 yo (34–68) 3 mixed		- Concordant Loss of BAF250a Expression in 14 EaOC and in 10 contiguous AE: 6 of the 10 cases (60%) showed BAF250a loss in the contiguous AE, and 3 of these 6 cases also demonstrated loss in adjacent TE as well - In benign ovarian endometrioma, all cases of AE showed retention of BAF250a also in TE.
Andersen et al. [47]	2018	Retrospective Study	E2sig, ERα, ERβ, PR	NR	11, 47 (range 34–20)	19, 57.5 (range 47–77) 5 CCC 14 EnOC	11, 39 (range 25–74)	- Decreasing ERα and PR protein from benign endometriosis to EAOC. - ERβ and FGF18 mRNA expression increases incrementally from benign endometriosis to EAOC. - Some ERα induced genes decrease during the progression of endometriosis to EAOC while several ERα-induced genes remain highly expressed.
Zhang et al. [48]	2018	Retrospective Study	genome-wide transcriptomic profiling	NR	4, NR	5 EnOC, NR	4, NR	- Distinct clustering between EnOC with and without concurrent endometriosis and AE. - NFkB, RAS, and TGF-b signaling are involved in EnOC associated with endometriosis and AE.
Niguez-Sevilla et al. [49]	2019	Prospective Study	Ki-67, BAF250a, COX-2	2014–2017	23	26	159	- The Ki-67 higher in AE than in BE. - Higher COX-2 expression in BE than in AE. - Higher loss of BAF250a expression in AE than BE.
Jiao et al. [50]	2019	Case Report	CK7, CEA, p16, CA 125, MUC-6, p53, PTEN, ER, PR, CK20, PAX-8, CDX-2	NR	46 yo	1 OMC	NR	- Tumor cells positive for CK 7, CEA, p16, CA125, MUC-6, and p53. They were negative for PTEN, ER, PR, CK19, CK20, PAX-8, and CDX2. - The atypical endometriotic epithelium was positive for PAX-8.
Păvăleanu et al. [51]	2020	Retrospective Study	E-cadherin,β-catenin, CK18, Bcl-2/Bax, ER, PR	2005-2017	--	19 (8 EnOC; 11 non EnOC), 59.10 yo	31, 36.61 yo	- Higher immunoexpression of CK18 and E-cadherin in endometriosis than in neoplastic counterparts. - β-catenin had stronger immunoexpression in tumors compared with endometriotic areas *. - Bcl-2/Bax higher rate in endometriosis had a statistically significant association with more aggressive tumor behavior *. - PR immunostaining correlated with ovarian location of endometriosis and tumor grade of EAOC *. - Stromal ER and PR immunoexpression were significantly lower in endometriosis in comparison to tumor stroma, and PR stromal immunoexpression had been higher in more differentiated tumors compared to less differentiated types *.
Penciu et al. [52]	2020	Case Series	ER, PR, Ki67, p53, p16, WT1, CD 34, CD10	2015–2017	--	2 EnOC, 2 OSC; 30–60 yo	--	- No similarities between endometriosis and ovarian cancer and no AE was identified. - Higher values of ER expression in endometriosis were associated with OSC than in those associated with EnOC. Higher values of ER expression were also recorded in ovarian cancer than in endometriotic foci. - An aberrant expression of p53 and p16 was noted only in OSC. - Positive WT1 was identified only in OSC.
Lenz et al. [53]	2021	Retrospective Study	ER, PR, Ki67, p53	NR	5, NR	--	40, 33.1 yo (22.0–47.0)	- In AE, higher and strong p53 expression (on average 26%) and decrease in ER (on average 56%) and PR (less than 1%) expression was observed *.
Shin et al. [54]	2021	Case control study	TSPAN1	NR	18 (40 ± 6.6)	7 AdjEm (44.5 ± 11) 17 CCC (43.6 ± 10.6) 12 EnOC (48.5 + 10)	9 (27.7 ± 5.4)	- Consistent increase in expression of all 14 genes from endometriosis to AE and AdjEm and finally to OCCC. - TSPAN1 expression is higher in early stage OCCC and AE. - TSPAN1 increases cell growth via AMPK phosphorylation in endometriosis cell lines (response to hypoxia). - TSPAN1 knockdown reduces OCCC cell growth via a mechanism not involving AMPK.

* *p*-values statistically significant (*p* < 0.05). Legend. NR, Not reported; AE, atypical endometriosis; TE, Typical endometriosis; EAOC, endometriosis-associated ovarian cancer; yo, years old; MSI, Microsatellite Instability; ER, Estrogens Receptor; PR, Progesterone receptor; EGFR, Epithelial Growth Factor Receptor; HNF-1beta, hepatocyte nuclear factor-1beta; LOH, Loss of Heterozygosity; PTEN, Phosphatase and tensin homolog; EnOC, ovarian endometrioid carcinoma; CCC, ovarian clear cell carcinoma; CCAF, benign clear-cell adenofibroma; OSC, ovarian serous carcinoma; OMC, ovarian mucinous carcinoma; E2sig, a panel of 236 genes associated with endocrine response, 158 of which showed statistically significant changes in expression between different disease states; FGF18, Fibroblast Growth Factor 18; L1-ICAM, L1, cell adhesion molecule; PDGFR, platelet-derived growth factor receptor; PDGF, platelet-derived growth factor; Cks, cyclin-dependent kinase subunit 1; p-mTOR, phosphorylated mammalian target of rapamycin; ARID1A, AT-rich interactive domain 1;AHIF-1, hypoxia-inducible factor-1; Glut1, glucose transporter 1; COX-2, cyclooxygenase-2; SRAP, steroid receptor RNA activator protein; ASC, adenosquamous carcinoma; AS, adenosarcoma; iNOS, inducible nitric oxide synthase; CEA, Carcinomatous embryonic antigen; CK, Cytokeratin; CA 125, Carbohydrate Antigen 125; MUC-6, Mucine 6; PAX-8, Paired-box gene 8; CDX2, Caudal Type Homeobox 2; Bcl-2, B-cell lymphoma 2; Bcl-2-associated X, BAX; CD, cluster of differentiation; WT1, Wilm’s tumor 1; TSPAN1, Tetraspanin1; AdjEm, adjacent endometriosis to OCCC; NE, normal endometrium; BE, benign endometriosis; C3, C4a, C7, CFD, CFB, CFH, Complement factors 3, 4a, 7, D, B, H;, MASP1, mannose-associated serine protease 1; IHC, immunohistochemistry; BAF250a, Brahma-associated factor (BRG-)associated factor 250a; HNF-1β, hepatocyte nuclear factor 1β; IMP3, insulin-like growth factor II mRNA-binding protein 3; PPV, Positive predictive value; NPV, Negative predictive value; PIK3CA, phosphatidylinositol-4,5-bisphosphate 3-kinase catalytic subunit alpha. * Nucleolar organizer regions (NORs) are loops of DNA, which are associated with nonhistone nucleoproteins, and they can be stained with silver (AgNORs). An increased number of AgNORs per nucleus is observed in many malignancies.

## Data Availability

The data presented in this study are available on request from the corresponding author.

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
