# Peer review of "A Systematic Review of Atypical Endometriosis-Associated Biomarkers"

_ijms, 2022, doi:10.3390/ijms23084425_

Round 1
Reviewer 1 Report
Review of the resubmitted article entitled „A systematic review of atypical endometriosis associated biomarkers”.
The article has gone through major revision before the resubmission. It is a systematic review focusing on known biomarkers atypical endometriosis and its association with ovarian cancer. It is a well structured review, and regarding atypical endometriosis it is stated to be the first in the subject, although there were previous papers on biomarkers of ovarian endometriosis and association with ovarian cancer (eg. Mikhaleva LM, Davydov AI, Patsap OI, Mikhaylenko EV, Nikolenko VN, Neganova ME, Klochkov SG, Somasundaram SG, Kirkland CE, Aliev G. Malignant Transformation and Associated Biomarkers of Ovarian Endometriosis: A Narrative Review. Adv Ther. 2020 Jun;37(6):2580-2603. doi: 10.1007/s12325-020-01363-5). Citation of this paper in the article would be useful.
The materials and methods part of the review process is valid.
Massive molecular evidence was collected in the past few years in terms of endometrial cancer, which had a major impact on guidelines and the therapy of the disease. Regarding the similarity in the genetic alterations in atypical endometriosis and endometriosis associated ovarian cancer, it would be useful to imply to these data in the discussion section of the article, since it may have a major future effect on the direction of research in upcoming studies.
The changes regarding my previous review opinion were made, and regarding the present form I do not have major suggestions except for the ones above.
Despite the results of the present review finds that there are no easily and cheaply detectable serum biomarkers regarding atypical endometriosis, and that the present evidence is from diverse histological and molecular based results based on low patient number studies, to my best opinion it is still a valuable and well structured review of the field. The results are not sufficient to change the clinical management of the disease, but provide a useful tool in summing the present available data in the field.
Author Response
Thank you for your kind suggestion. We added the reference in the text (Mikhaleva et al) and we imply data regarding common genetic mutations between endometriosis-associated ovarian cancer and atypical endometriosis in the discussion section.
Reviewer 2 Report
I appreciate the authors' effort to re-structure this manuscript. This manuscript had been greatly improved and was easier to read. I only have minor comments for the authors.
1. Please spread out the following abbreviations, AE, BE, CCC, EAOC, EnOC, and TE, throughout your manuscript to further improve the readability of your study.
2. In the 3.1 section- Please define the abbreviation TE in this section. Otherwise, I suggest spread out TE throughout your manuscript.
Author Response
Thank you for your kind suggestion. We correct all the abbreviations in the text.
Reviewer 3 Report
The authors addressed my comments and I recommend the publication of the manuscipt in the present form
Author Response
We really appreciate the reviewer's comment.
Reviewer 4 Report
Dear Authors,
Generally, I like your manuscript and think that your topic is actual. However, some corrections will be required.
The Introduction section is too short. In my opinion, you should add more citations concerning the theories and statistics. It always helpful in some kind of clinical reviews.
The same suggestion for 3.5 "growth factors and their receptors". Not enough data. Also, epigenetic changes - I have not seen this data concerning endometriosis in the current review.
Thank you.
Author Response
Thanks to the reviewer for the suggestion. We improved the introduction as suggested by the reviewer. About the "Growth factor" section in the results paragraph, we agreed with the reviewer regarding the few data available, but, unfortunately, only 4 studies are published on the issue. We just reported the main data, this is the reason why we choose not to face the argument in the discussion section. Finally, no studies assessing the relationship between AE and epigenetic factors have been reported to our knowledge, so we cannot include this aspect in the systematic review.
This manuscript is a resubmission of an earlier submission. The following is a list of the peer review reports and author responses from that submission.
Round 1
Reviewer 1 Report
In this article, the authors performed a retrospective and systematic review about the possible markers in pre-malignant ovarian endometriosis and endometriosis-related ovarian cancer. The authors propose the neutrophil-to-lymphocyte ratio as possible inflammatory marker in atypical endometriosis. The manuscript is well written and of high interest for the endometriosis community. I recommend the publication of the article after addressing the following minor revisions:
- In the Abstract section, some abbreviations are used without spelling them out, for example, AE, OMA, PI3K.
- In general, some abbreviations are not spelled out when they are used for the first time but later. Some examples are: BE (at the end of the Introduction section), OR, aPTT, ASRM, r-AFS, among others. Moreover, sometimes the terms are abbreviated and sometimes are not. I recommend extensively checking all the abbreviations in all the text and unifying their use.
- At the beginning of page number 2, “It has been estimated that 0.5-1% of cases are complicated by neoplasia, with a lifetime risk…”, to which cases are the authors referring to? Endometriosis? Atypical endometriosis? Please, clarify.
- In the Materials and Methods section, when describing the Systematic Review of the Literature, “clarity group” should be written with capital letters “Clarity group”.
- In Materials and Methods section, when describing the assessment of the risk of bias for each study, Supplementary Figure 1 should be mentioned (in page 3 or 4).
- In Table 3, in the description of abbreviations, OC and PLR are spelled out although they are not used in the table.
- In table 4, please, describe what “yo” means.
- This work focuses on the description of different possible biomarkers for atypical endometriosis, therefore the characteristics of atypical endometriosis and differences with typical endometriosis should be more extensively explained for the reader. I recommend describing it in the Introduction section.
- I recommend adding a table describing the characteristics that define the different ovarian cancer mentioned in the text (CCC, endometrioid adenocarcinoma, serous carcinomas, adenosarcoma, etc.).
Reviewer 2 Report
This study aimed to we examined whether atypical endometriosis has an increased NLR and PLR expression compared to benign endometriosis and endometriosis associated ovarian cancer. This study was interesting, however, I think the low number of patients did not have statistical power to support the conclusions. The designs of study may also bear flaws. The authors may need to re-organize the design of this study. In addition, this manuscript was poor-written and contained too many abbreviations (e.g. AE, BE, OMA, typical OMA, or atypical OMA) that lack consistency throughout this manuscript. I had difficulty in reading this manuscript and can't easily follow. Hence, specific comments can not be provided at current situation. I suggest the authors to re-organize and re-write this manuscript to meet the standard of SCI journal.
Reviewer 3 Report
The review part of the article goes through the known biomarkers that can differentiate between endometriosis and EOC, with explanation on the genetics of the disease as well. The review part of the article is well written and gives an up- to-date overview of the subject. Some parts e.g.: “The same mutation was retrieved in 6/8 non-atypical endometriotic lesions (75%) and 7/8 atypical endometriotic lesions (88%). These data provide evidence that mutations of the PIK3CA gene occur in the putative precursor lesions of CCC…” draw far conclusions from the molecular classification of just 8 cases.
There are a few typing errors in the paper. (e.g. „P” is not capital, „Atypical” endometriosis is not with capital letter, OMA abbreviation should be explained in the abstract).
The case – control part of the paper is more problematic, and needs full revision:
The authors say it was a retrospective case- control study based on a retrospective analysis on a surgical database collected between 2015-2021. Yet it is written that blood was drawn from each patient one month before the operation. Was this the blood sample taken routinely for the preoperative investigation? They also say all patients had signed written informed consent. Was this informed consent signed retrospectively if this was a retrospective analysis?
According to the 2018 revised FIGO staging of ovarian cancer there is no such thing as IIC stage. (This is even true to the 2014 FIGO staging system). There would be no point to use an older staging system in the study.
There is a significant difference in size of the cysts between endometriosis and ovarian cancer, also between the age of the groups, the latter patients being in postmenopause, therefore this is not a correctly matched case- control group. The problem is that postmenopausal endometriosis carries out a well known risk for ovarian cancer (eg. the ROMA score of these patients should have been very high for the postmenopausal most probably elevated CA125), and the size of the ovarian cysts over 8 or 10 cm.-s is a risk factor according to most ultrasound prediction models as well. If this is a retrospective analysis a new control group should be selected which does not differ significantly in age and size from the OC group of patients, because this is a very strong bias. In the charts it is not clear that the statistical significance applies only to results between endometriosis without atypia and EOC. This gets clear only from the text below. Do we have literature information about the change of the studied ratios with age and menopausal status? The levels and sensitivity of HE4 and CA125 markers change with postmenopause significantly, respectively. These are both markers widely used in the detection of ovarian cancer and the correlation of the latter one with endometriosis is also known.
The authors say no statistical difference was observed between atypical OMA- typical OMA, and atypical OMA- EOC in regards of parameters NC. LC. NLR. However, the sample size of atypical OMA was only 15, the other 2 was 45, therefore it is questionable if this non significance was due to the small sample size of atypical OMA.
The solution to the above is finding a matching control group for the EOC patients, without that the study cannot be evaluated. Also, the number of atypical OMA patients should be increased to draw any conclusion in that direction, or specific statistical tests should be presented about the sufficiency of the sample size. I only suggest accepting the paper if statistical significance is proven with a correct matching control group as well.
Reviewer 4 Report
This manuscript is a mixture of original research and a systematic review. While I believe both works have merit they need to be separated in unique enteties and treated as such.
The original research connected to the "Neutrophil-to-Lymphocyte and Platelet-to-Lymphocyte ratios in pre-malignant endometriotic lesions and endometriosis-associated-ovarian-cancer" needs to be elaborated further and discussed in detail in order to be an addition to the current state of the art.
The systematic review presented in the manuscript itself is too broad in the field of atypical markers and too little interconnected to the topic of neutrophil-to-lymphocyte and platelet-to-lymphocyte ratio. Therefore a quality judgement of merit is questionable until these two entities are not separeted in 2 works.